

# Evolution of the climatic tolerance and postglacial range changes of the most primitive orchids (Apostasioideae) within Sundaland, Wallacea and Sahul

Marta Kolanowska[1,2,*], Katarzyna Mystkowska[1], Marta Kras[1], Magdalena Dudek[1] and Kamil Konowalik[3,*]

[1] Department of Plant Taxonomy and Nature Conservation, University of Gdansk, Gdańsk, Poland
[2] Department of Biodiversity Research, Global Change Research Institute AS CR, Brno, Czech Republic
[3] Department of Plant Biology, Institute of Biology, Wrocław University of Environmental and Life Sciences, Wrocław, Poland
[*] These authors contributed equally to this work.

## ABSTRACT

The location of possible glacial refugia of six Apostasioideae representatives is estimated based on ecological niche modeling analysis. The distribution of their suitable niches during the last glacial maximum (LGM) is compared with their current potential and documented geographical ranges. The climatic factors limiting the studied species occurrences are evaluated and the niche overlap between the studied orchids is assessed and discussed. The predicted niche occupancy profiles and reconstruction of ancestral climatic tolerances suggest high level of phylogenetic niche conservatism within Apostasioideae.

## INTRODUCTION

The orchid subfamily Apostasioideae includes only two genera, *Apostasia* and *Neuwiedia* which are restricted in their distribution to Asia and Australia. They are small to medium-sized terrestrial plants producing elongated, fibrous roots. Their leathery, plicate leaves are spirally arranged, spreading or suberect. Inflorescences are terminal, racemose, few- to many-flowered. Flowers are actinomorphic to slightly zygomorphic, with lip similar to petals or slightly broader (*Xinqi, Gale & Cribb, 2009*). In apostasioid orchids the column part is usually obscure. Two or three fertile anthers, representing the median of the outer whorl, and the laterals of the inner whorl are present. All stigma lobes are similar to each other in size and shape. Pollen grains do not form any pollinia (*Szlachetko & Rutkowski, 2000*).

Apostasioideae have been intensively studied in the contexts of their taxonomic affinities (*Kocyan & Endress, 2001*), phylogenetics (*Judd, Stern & Cheadle, 1993*; *Kocyan et al., 2004*), anatomy (*Stern, Cheadle & Thorsch, 1993*), mycorrhizal associations (*Yukawa et al., 2009*)

Corresponding author
Marta Kolanowska,
martakolanowska@wp.pl

and genome size (*Jersáková et al., 2013*). So far, however, no research on their biogeography has been conducted.

The divergence between Apostasioideae and other orchids occurred in the Mesozoic. The age of ancestor of this subfamily is estimated at 41–49 Ma (*Ramírez et al., 2007*; *Gustafsson, Verola & Antonelli, 2010*) which was a time of important geological events. The occurrence of apostasioids has been reported from India, Nepal, Bhutan, Japan, south-east Asia, New Guinea and northern Australia (*Xinqi, Gale & Cribb, 2009*). Their geographical range significantly overlaps one of the least-understood regions of the world. The enormous geological activity of this area has created a complex, fragmented pattern of numerous islands. During the frequent past glacial periods numerous islands were connected by dry land forming two large masses called Sundaland and Sahul. These two areas are separated by Wallacea, a group of mainly Indonesian islands which were never bridged by dry land during glaciation (*O'Connell, Allen & Hawkes, 2010*). Research by *Van Hinsbergen et al. (2012)* indicated that a collision of an extended microcontinental fragment and continental Asia occurred approximately 50 Ma, and it was followed by a continent to continent collision about 25 Ma. The dispersal of Indian floristic elements in Southeast Asia estimated based on palynological data (*Morley, 1998*) took place in the middle Eocene (50–39 Ma). Moreover, in the late Eocene, the Tasmanian Gateway opened (*Exon et al., 2002*). The paleogeography varied with eustatic sea level fluctuations, however, from the Eocene to Early Miocene most of Sundaland was terrestrial with volcanic areas at the southern margin. Australia began to collide with south-eastern Sundaland about 23 million years ago closing the former deep ocean separating the two continents (*Hall, 2013*). The Neogene history of the SE Asian gateway is a complex history of rapid changes in tectonics, topography and land/sea distributions (*Hall, Cottam & Wilson, 2011*). Simulations of the paleoclimate at the Last Glacial Maximum (LGM; about 22,000 years ago) indicated that the majority of Sundaland was well within climatic envelope for tropical everwet forest (*Cannon, Morley & Bush, 2009*).

As mentioned before, biogeographical research on Apostasioideae, especially those on their history, were not published. The deficiencies in the knowledge of the distribution of glacial refugia and further migration concern numerous herbaceous tropical plants which are often absent in fossil material. In such cases, the analysis of postgalcial history is usually based on molecular analyses (e.g., *Nordström & Hedrén, 2008*). However, the results of these research are not always consistent with the distribution of their suitable climatic niches in glacial and postglacial period (*Naczk & Kolanowska, 2015*). For tropical plants, it is also difficult to estimate their ecological tolerances for climatic factors based on traditional field observations as well as to select crucial variables limiting their occurrence. For that reason, the ecological differences between numerous closely related species remain unknown.

The aim of the present study was to combine results of classical herbarium studies with modern computer tools and molecular data to reconstruct postgalcial history of Apostasioideae representatives and to evaluate variation in their preferred climatic niches. It is particularly important for endangered plants, since the knowledge of their past may be a guidance how the organisms will adapt in response to global climate change in the future.

The recognition of the niche conservatism within closely related species is also essential for understanding their evolution.

Our research focused on estimation of the location of possible glacial refugia of studied orchids during the LGM and to evaluate postglacial changes in the distribution of their ecological niches using ecological niche modeling (ENM) analysis. The results of ENM were combined with the phylogenetic study to reconstruct ecological tolerances of the studied species ancestors and to visualize evolution of the climatic tolerances within Apostasioideae. Additionally, the differences between climatic niches occupied by representatives of *Apostasia* and *Neuwiedia* and climatic factors limiting their occurrence were evaluated.

## MATERIAL AND METHODS

### Data sources

A database of the sites of Apostasioideae representatives was prepared based on the examination of herbarium specimens stored in BM, K, MO, and P, as well as the electronic databases of AAU, BRI, CANBR, CNS, NOCC, SCBI, SING, US and the Swiss Orchid Foundation. The herbaria acronyms follow Index Herbariorum (*Thiers, 2015*). Only the sites that could be precisely placed on the map were used in our analyses. The process of georeferencing followed *Hijmans et al. (1999)*. The geographic coordinates provided on the herbarium sheet labels were verified. If there was no georeferencing information on the herbarium sheet label, we followed the description of the collection place and assigned coordinates as precisely as possible.

We did not find enough material (occurrence data) to conduct analysis for *Apostasia latifolia* Rolfe, *Apostasia parvula* Schltr., *A. ramifera* S.C. Chen & K.Y. Lang, *A. shenzhenica* Z.J. Liu & L.J. Chen, *Neuwiedia elongata* de Vogel, *N. griffithii* Rchb.f., *N. inae* de Vogel, *N. malipoensis* Z.J. Liu, L.J. Chen & K. Wei Liu, or *N. siamensis* de Vogel. We gathered a total of 179 occurrence records of three *Apostasia* and three *Neuwiedia* species (Table S1): 27 of *Apostasia nuda* R. Br., 22 of *A. odorata* Blume, 80 of *A. wallichii* R. Br., eight of *Neuwiedia borneensis* de Vogel, 24 of *N. veratrifolia* Blume, and 18 of *N. zollingeri* Rchb. f. To reduce sample bias we applied spatial filtering (*Boria et al., 2014*). From a total of 179 records, we randomly removed sites of each species that were within 25 km of one another. The final database included 13 sites of *Apostasia nuda*, 13 of *A. odorata*, 41 of *A. wallichii*, four of *Neuwiedia borneensis*, 18 of *N. veratrifolia*, and 10 of *N. zollingeri*.

### Ecological niche modeling

The ecological niche modelling was conducted using the maximum entropy method implemented in MaxEnt version 3.3.2 (*Phillips, Dudik & Schapire, 2004*; *Phillips, Anderson & Schapire, 2006*; *Elith et al., 2011*) based on the species presence–only observations. From 19 climatic variables ("bioclims", Table 1) in 2.5 arc minutes ($\pm 20$ km2 at the equator) developed by *Hijmans et al. (2005)* we removed seven "bioclims" due to their significant correlation (above 0.9) as evaluated by the Pearson's correlation coefficient calculation computed using ENMTools v1.3. The following variables were excluded from the dataset: bio6, bio7, bio9, bio10, bio11, bio16 and bio17. As it was suggested that using a restricted area in ENM analysis is more reliable than calculating habitat suitability on the global

**Table 1  Bioclimatic variables used in the ENM analysis.**

| Code | Variable |
|------|----------|
| bio1 | Annual mean temperature |
| bio2 | Mean diurnal range = Mean of monthly (max temp–min temp) |
| bio3 | Isothermality (bio2/bio7) (*100) |
| bio4 | Temperature seasonality (standard deviation *100) |
| bio5 | Max temperature of warmest month |
| bio8 | Mean temperature of wettest quarter |
| bio12 | Annual precipitation |
| bio13 | Precipitation of wettest month |
| bio14 | Precipitation of driest month |
| bio15 | Precipitation seasonality (coefficient of variation) |
| bio18 | Precipitation of warmest quarter |
| bio19 | Precipitation of coldest quarter |

scale (*Barve et al., 2011*), the area of our analysis was restricted to longitude 88°–190°E and latitude 53°S–47.8°N. The maximum iterations was set to 10,000 and the convergence threshold to 0.00001. The "random seed" option which provided the random test partition and background subset for each run was applied. The run was performed as a bootstrap with 1,000 replicates, and the output was set to logistic. The analogical settings were used in the modelling for LGM. The bioclimatic data for this time period were developed and mapped (CCSM4) by Coupled Model Intercomparison Project Phase 5 (CMIP5; *Taylor, Stouffer & Meehl, 2012*). All operations on GIS data were carried out on ArcGis 9.3 (ESRI).

The evaluation of the models was done using the most common metric—area under the curve (AUC; *Mason & Graham, 2002*) which was calculated by the Maxent application automatically. The niche overlap between the studied species was calculated using ENMTools v1.3 (*Warren, Glor & Turelli, 2010*).

## Evolution of climatic tolerance and predicted niche occupancy (PNO)

To visualize niche evolution for Apostasioideae the ancestral tolerances were pictured on a phylogenetic tree. To construct the tree for taxa included in the ecological niche modeling sequences from ITS, *mat*K and *trn*L available from GenBank (Table S2) were aligned in Mafft 6.833b (*Katoh & Toh, 2008*) and Gapcoder (*Young & Healy, 2003*) was used to code indels. Aligments were merged and a Bayesian phylogenetic analysis was performed in MrBayes 3.2.1 (*Ronquist et al., 2012*). In the nucleotide part for each marker separately, a model from the best selection according to AIC implemented in jModelTest 2.1.1 (*Darriba et al., 2012*) was used. For the binary coded gaps, a Jukes–Cantor model (*Jukes & Cantor, 1969*) was used. 15,000,000 generations were performed in two runs discarding the first 25% as the burnin fraction and sampling every 1000th tree. As an outgroup *Cypripedium subtropicum* was used. To estimate node ages, function chronos in package "ape" was used (*Paradis, Claude & Strimmer, 2004*) applying lambda set to 20. In this function, branch lengths elucidate mean numbers of substitutions and node ages are estimated using semi-parametric method based on penalized likelihood

(*Sanderson, 2002*; *Paradis, Claude & Strimmer, 2004*). As calibration points splits between Apostasioideae and *Cypripedium* (80 Mya), and *Apostasia* and *Neuwiedia* (43 Mya) were used following published divergence times (*Gustafsson, Verola & Antonelli, 2010*). To reconstruct ancestral ecological tolerances and predicted niche occupany profiles (PNOs) the Phyloclim package was used (*Heibl & Calenge, 2013*) which implements the methodology originally developed by *Evans et al. (2009)*. PNO profile takes into account species probability distribution derived from ecological niche modeling and compiles a response to a particular environmental variable for each species. Ancestral ecological tolerances are computed from the phylogenetic tree and PNO using nonparametric approach and ancestral character estimation (*Paradis, Claude & Strimmer, 2004*; *Evans et al., 2009*).

# RESULTS

## Models evaluation

All the created models received high AUC scores of 0.898–0.989. These results are consistent with the outcomes of previous studies which indicated the reliable performance of this method for developing ecological niche models based exclusively on presence-only data (*Elith et al., 2006*).

## Glacial refugia

The ENM analysis indicated eastern Mindanao, Borneo, Sumatra, eastern Sunda, and northern Sahul as areas suitable for the occurrence of *Apostasia nuda*. The potential refugia of *A. odorata* are widely distributed along the coast of Sahul and Sunda, as well as between those shelves. The highly suitable niches of *Apostasia wallichii* were limited to the north-eastern part of Sunda. The created models indicated an even smaller potential coverage area of the glacial refugia of *Neuwiedia borneensis*. The most suitable niches were limited to numerous small islands north-east of Sahul. Some less appropriate habitats could be found in Borneo and south of this region, as well as along northern Sauhl. The refugia of *N. veratrifolia* were widely distributed in Sunda, Sumatra, Java, Mindanao, and northern Sahul. Suitable niches of *Neuwiedia zollingeri* during LGM were located in islands west of Sunda, eastern Mindanao, Sulawesi and smaller islands north-east of Sahul. Figures 1 and 2.

## Current potential range

Currently, the most suitable niches of *Apostasia nuda* are located in Peninsular Malaysia, Borneo, Sumatra, western Mindanao, Sulawesi, Halmahera, north-western New Guinea and New Britain. The potential range of *A. odorata* is very wide within the studied area. No suitable climatic conditions were indicated by the created models in the areas of higher elevation in continental south-east Asia and the central part of New Guinea. In Australia the appropriate niches are located in Tasmania, in Cape York Peninsula, and along the northern, south-eastern and south-western coasts. Some areas characterized by the climatic conditions suitable for this species are also located in New Zealand. The potential range of *A. wallichii* includes the coasts of Thailand, Cambodia and Vietnam. It may also occur
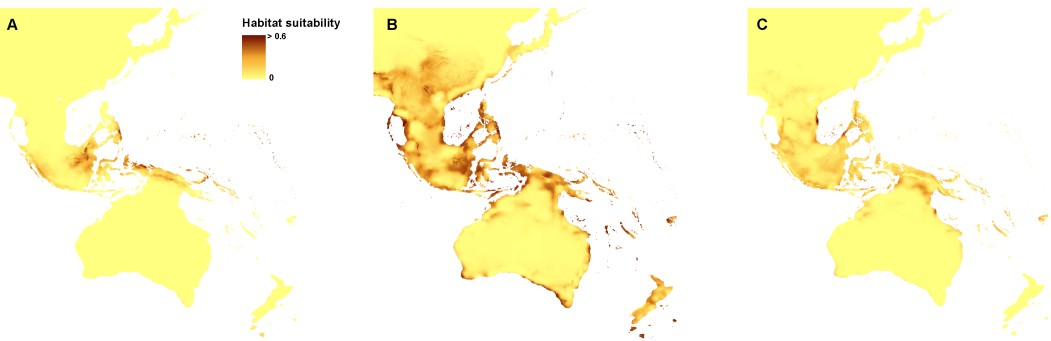

**Figure 1** **Distribution of the suitable climatic niches of *Apostasia* species during LGM: *A. nuda* (A), *A. odorata* (B), and *A. wallichii* (C).** Map was prepared by the authors using ArcGIS software.

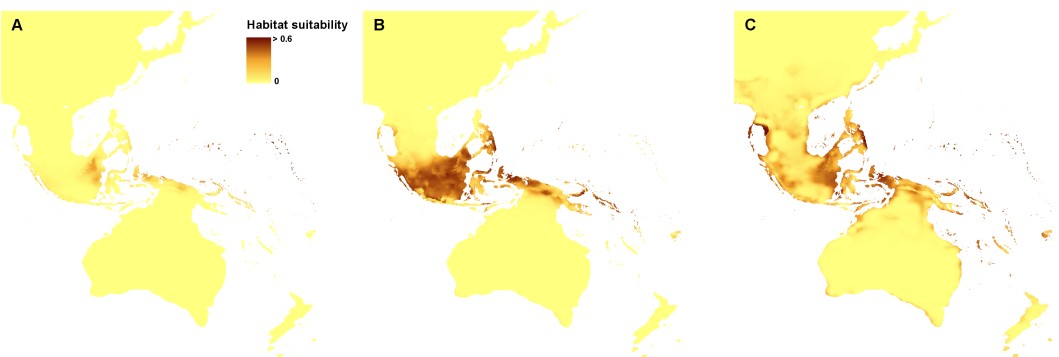

**Figure 2** **Distribution of the suitable climatic niches of *Neuwiedia* species during LGM: *N. borneensis* (A), *N. veratrifolia* (B), and *N. zollingeri* (C).** Map was prepared by the authors using ArcGIS software.

in Burma, southern Sumatra, Peninsular Malaysia, the Greater Sunda Islands, Borneo, Sulawesi, Muluccas, New Guinea, Cape York Peninsula, the Solomon Islands, Vanuatu, New Caledonia, and Fiji. Suitable niches of *N. borneensis* are located in Borneo, northern New Guinea, and Halmahera. Areas characterized by less appropriate climatic conditions are also found in Sumatra and the Solomon Islands. The potential range of *N. veratrifolia* includes Peninsular Malaysia, northern Sumatra, Palawan, Mindanao, Sulawesi, Moluccas, central New Guinea, and the Solomon Islands. Some less suitable niches are located in Borneo, Java, and Vanuatu. Habitats of *N. zollingeri* are distributed in Laos, Vietnam, Peninsular Malaysia, the Nicobar Islands, Sumatra, Borneo, the Philippines, Sulawesi, Moluccas, New Guinea, the Solomon Islands, Vanuatu, New Caledonia, and Fiji, as well as on the northern coast of Australia, and the Solomon Islands. Figures 3 and 4.

## Limiting factors and niches overlap

The distribution of all the studied species is related mostly with the mean diurnal range (Table 2). The second important climatic factor limiting the occurrence of apostasioid orchids is the precipitation in the warmest quarter, which significantly influenced the models of the three *Apostasia* species and *Neuwiedia borneensis*. The geographical ranges of *Neuwiedia veratrifolia* and *N. zollingeri* are related with isothermality and the annual mean
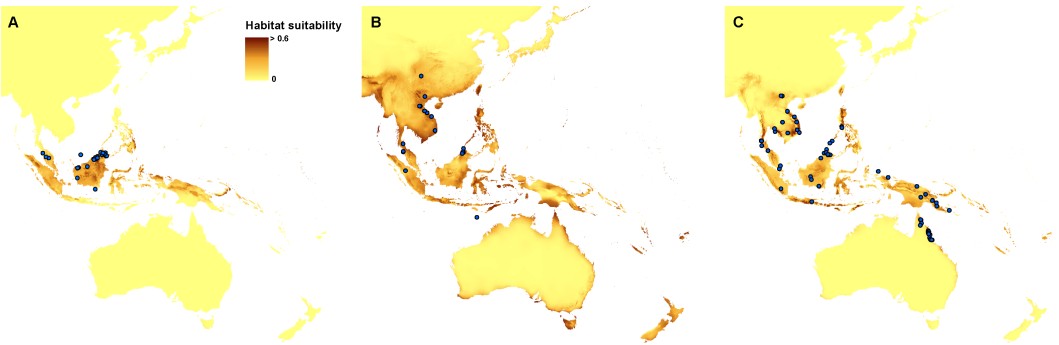

**Figure 3** **Current distribution of the suitable climatic niches of *Apostasia* species: *A. nuda* (A), *A. odorata* (B), and *A. wallichii* (C).** Spots indicate the localities used in ENM analysis. Map was prepared by the authors using ArcGIS software.

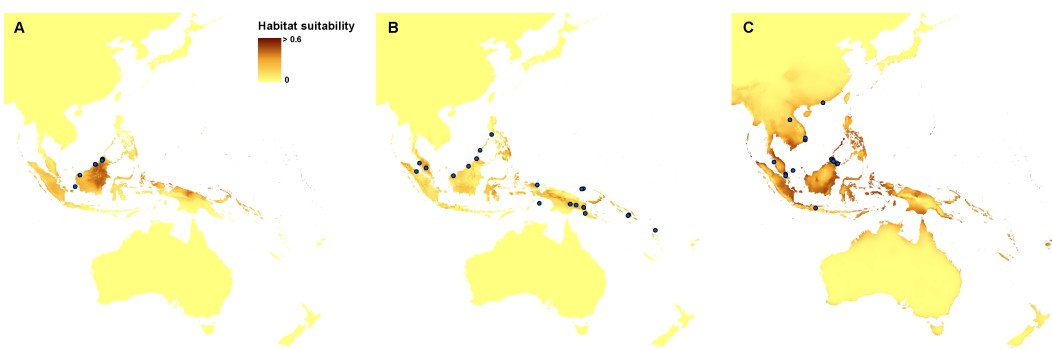

**Figure 4** **Current distribution of the suitable climatic niches of *Neuwiedia* species: *N. borneensis* (A), *N. veratrifolia* (B), and *N. zollingeri* (C).** Spots indicate the localities used in ENM analysis. Map was prepared by the authors using ArcGIS software.

**Table 2** **Estimates of relative contributions of the three most important environmental variables to the Maxent models.**

|       | *A. nuda*    | *A. odorata* | *A. wallichii* | *N. borneensis* | *N. veratrifolia* | *N. zollingeri* |
|-------|--------------|--------------|----------------|-----------------|-------------------|-----------------|
| var_1 | bio2 (35.1)  | bio2 (35.1)  | bio2 (35.1)    | bio2 (35)       | bio2 (24.7)       | bio2 (24.7)     |
| var_2 | bio18 (12.2) | bio18 (12.2) | bio18 (12.2)   | bio18 (12.2)    | bio1 (13.7)       | bio1 (13.7)     |
| var_3 | bio1 (10.5)  | bio1 (10.5)  | bio1 (10.5)    | bio1 (10.5)     | bio3 (11.5)       | bio3 (11.5)     |

temperature. The latter variable has a somewhat less significant contribution to the models of four other orchids.

The predicted niche occupancy graphs (Fig. 5) showed that while most of the analyzed climatic variables had a similar influence on the distribution of the studied species, some differences are observed in preferences related to precipitation: the precipitation of the driest month (bio14), precipitation seasonality (bio15), and the precipitation of the coldest quarter (bio19). *Apostasia* species grows in regions with low precipitation values recorded in the driest month, while *Neuwiedia* representatives prefer rainfall of about 100–130 mm at this time of year. In the aspect of precipitation seasonality, the value of about 25 is preferred by most of the studied species. However, three studied *Apostasia* representatives may also

**Table 3** Results of niche overlap tests calculated for LGM.

| D\I | A. nuda | A. odorata | A. wallichii | N. borneensis | N. veratrifolia | N. zollingeri |
|---|---|---|---|---|---|---|
| A. nuda | x | 0.627 | 0.689 | 0.940 | 0.891 | 0.783 |
| A. odorata | 0.349 | x | 0.849 | 0.560 | 0.670 | 0.924 |
| A. wallichii | 0.410 | 0.603 | x | 0.637 | 0.799 | 0.917 |
| N. borneensis | 0.731 | 0.311 | 0.367 | x | 0.816 | 0.736 |
| N. veratrifolia | 0.641 | 0.398 | 0.518 | 0.527 | x | 0.832 |
| N. zollingeri | 0.528 | 0.704 | 0.710 | 0.475 | 0.600 | x |

**Table 4** Results of niche overlap tests calculated for the present time.

| D\I | A. nuda | A. odorata | A. wallichii | N. borneensis | N. veratrifolia | N. zollingeri |
|---|---|---|---|---|---|---|
| A. nuda | x | 0.548 | 0.730 | 0.966 | 0.912 | 0.712 |
| A. odorata | 0.277 | x | 0.824 | 0.520 | 0.571 | 0.931 |
| A. wallichii | 0.468 | 0.567 | x | 0.707 | 0.793 | 0.896 |
| N. borneensis | 0.823 | 0.272 | 0.440 | x | 0.854 | 0.702 |
| N. veratrifolia | 0.678 | 0.315 | 0.541 | 0.577 | x | 0.728 |
| N. zollingeri | 0.463 | 0.715 | 0.663 | 0.456 | 0.485 | x |

grow in areas characterized by higher scores of this factor (about 80–85). While in the coldest quarter *Apostasia* species prefer low amounts of rainfall, *Neuwiedia* representatives are adapted to higher precipitation. The niches occupied by the studied *Apostasia* species did not overlap significantly during LGM (Table 3). *A. wallichii* and *A. odorata* received the highest scores of $D = 0.603$ and $I = 0.849$. In the glacial period, *Neuwiedia* representatives shared slightly more habitats. The highest overlap ($D = 0.600$, $I = 0.832$) was calculated for *N. zollingeri* and *N. veratrifolia* (Table 3). While within *Apostasia* the same pattern of niche commonality is observed in the present time, the habitat overlap within *Neuwiedia* has changed (Table 4). Currently, the highest niche overlap is observed for *N. borneensis* and *N. veratrifolia* ($D = 0.577, I = 0.854$).

Interestingly, the tests indicated that the potential ranges of several of the studied species overlap more significantly with representatives of other genera than their closest relatives. During LGM *N. borneensis* and *N. veratifolia* shared most of their potential range area with *A. nuda* and *N. zollingeri* with *A. odorata*. The same pattern is also observed at the present time. The overlap of the suitable niches of some species has decreased since LGM, e.g., *A. odorata* and *A. nuda*, *A. nuda* and *N. zollingei*, *A. wallichii* and *A. odorata*, *N. borneensis* and *A. odorata*, *N. veratifolia* and *A. odorata*, *N. veratifolia* and *A. wallichii*, *N. zollingeri* and *A. wallichii*. In other cases the potential ranges are currently more similar, e.g. *A. wallichii* and *A. nuda*, *A. nuda* and *N. borneensis*, *A. nuda* and *N. veratifolia*, *N. zollingeri* and *A. odorata*, *N. borneensis*–and *A. wallichii*.

## Evolution of climatic tolerance and predicted niche occupancy (PNO)

In this study the intra-specific variation was not addressed as it would require denser sampling throughout the whole region. However, when accepting taxonomical backbone

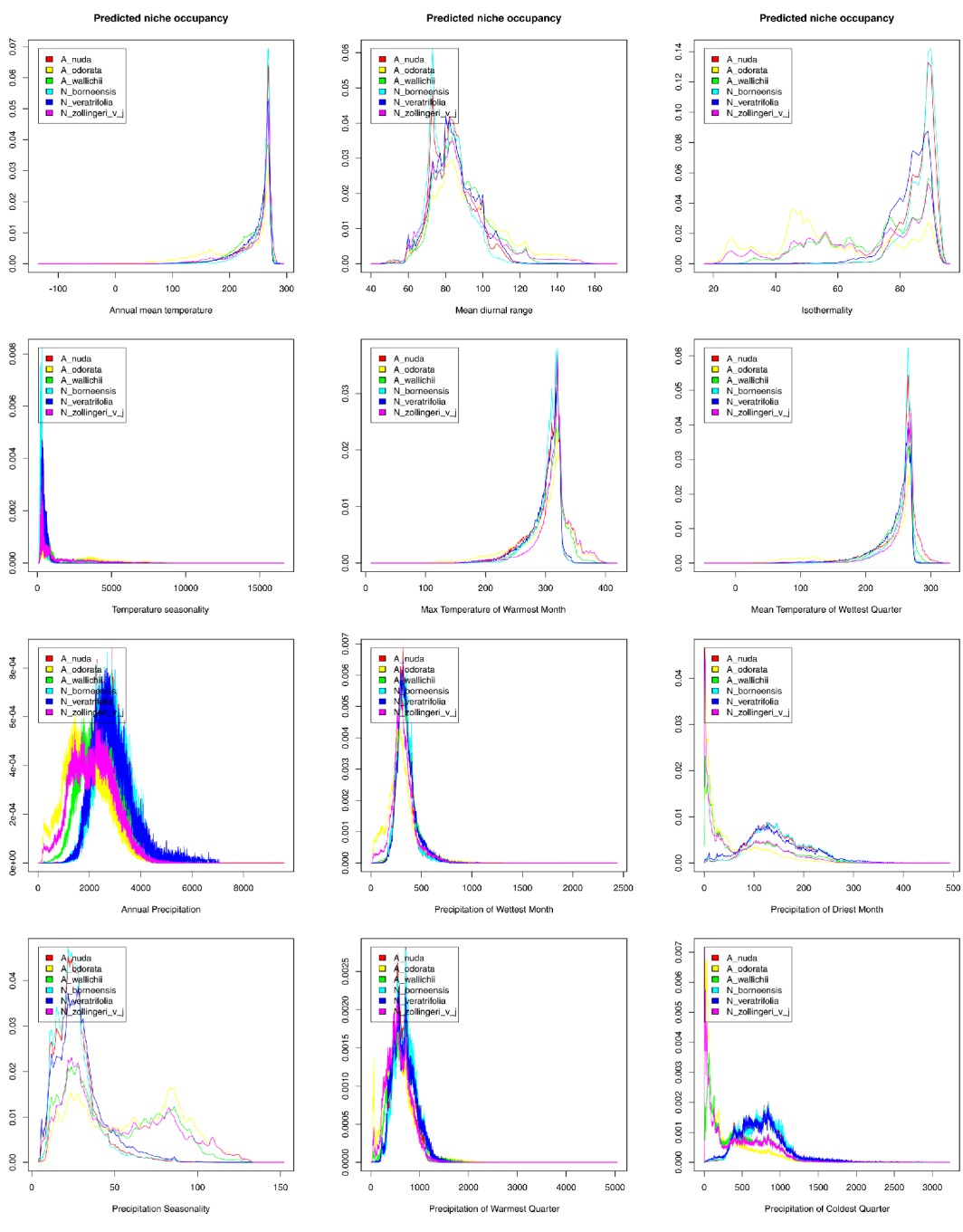

**Figure 5   Predicted niche occupancy profiles of all studied species.**

and treating species as being monophyletic, the assumption is made that there is no potential gene flow among taxa or possible diverse adaptations that are present within the intraspecific lineages. This may lead to some bias but currently available phylogenetic studies (*Judd, Stern & Cheadle, 1993*; *Kocyan et al., 2004*) suggest that it should be of minimal

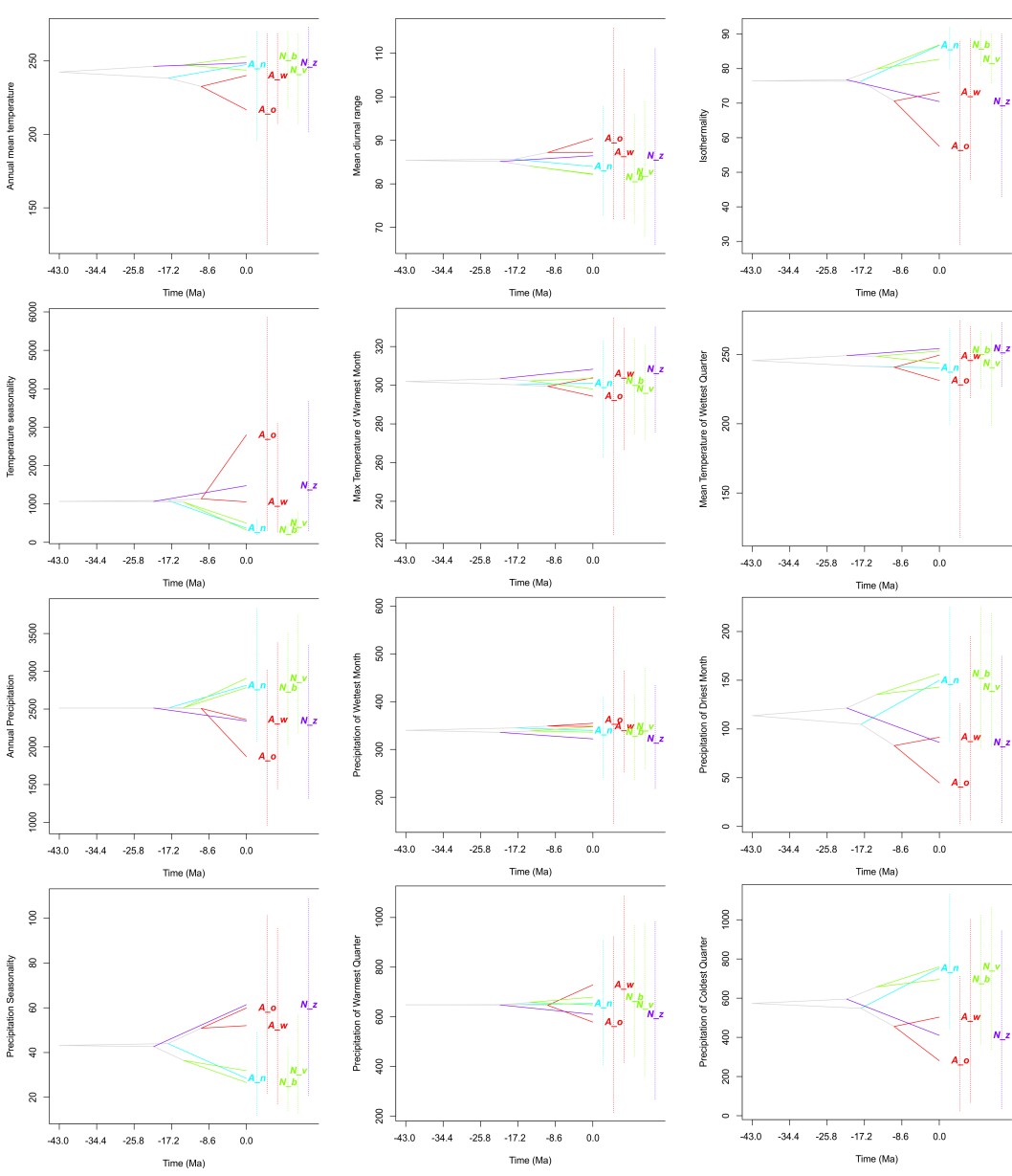

**Figure 6** **Evolution of ecological tolerances of all studied species.** *Apostasia wallichii*, $A_w$; *Apostasia odorata*, $A_o$; *Apostasia nuda*, $A_n$; *Neuwiedia zollingeri*, $N_z$; *Neuwiedia veratrifolia*, $N_v$; *Neuwiedia borneensis*, $N_b$.

importance since there is no gene flow between species nor any visual signs of local adaptations. The phylogenetic tree with marked estimated divergence times is presented in Fig. S1.

The evolution of climatic tolerance (Fig. 6) of all variables within the studied taxa indicated that *Apostasia odorata* developed somewhat broader ecological tolerance for four analyzed variables (bio1, bio5, bio8, bio13), in the case of another four (bio2, bio3, bio4, bio12) two species, *A. odorata* and *Neuwiedia zollingeri*, have distinctly broader tolerance. In relation to several climatic factors (bio14, bio18, bio19) all the studied species
present similar tolerance. It is noteworthy that *A. nuda* shows similar preferences in the aspect of many climatic variables (bio2, bio3, bio4, bio12, bio14, bio15 and bio19) to *N. borneensis* and *N. veratifolia*. In most cases, the tolerance of the studied species' ancestors diverged a long time ago into two lineages: the first included *Apostasia* species and the other *Neuwiedia*, and among these two groups the diversification of their ecological tolerances began relatively recently. Furthermore, the presented predicted niche occupancy profiles show that most species have very similar climatic requirements in respect to the studied variables, and deviations may be noted only in a few cases (Fig. 5).

## DISCUSSION

### Postglacial range changes

The ENM analysis did not indicate any significant changes in the distribution of the suitable niches of most of the studied taxa. The highest congruence is observed in the potential glacial and current ranges of *Neuwiedia borneensis* and *Neuwiedia zollingeri*. ENM analysis indicated a minor loss of areas characterized by suitable niches in *Apostasia odorata* and *Neuwiedia veratrifolia*. A slight extension of the potential range is observed in *Apostasia nuda* and *A. wallichii*. The second species probably occupied niches of lower suitability during the LGM.

Information provided in the atlas of palaeovegetation (*Adams & Faure, 1997*) indicated that all the studied species occupied areas covered by tropical rainforest and monsoon or dry forest during the LGM. *Apostasia odorata* and *A. wallichii* could have also occurred in relatively low, usually deciduous tropical woodlands. Potential glacial refugia of *Apostasia nuda* and *Neuwiedia veratrifolia* were also located in savannas. *Apostasia odorata* could also have survived the glacial period in montane tundra, grassland and semi-arid temperate woodlands or scrublands. The available data on the current distribution of the studied species are consistent with the habitat preferences during the LGM, and no significant shift is observed in any of these orchids. This would partially explain the lack of extensive migration after the glacial period.

### Potential vs documented geographical ranges

The potential range of the studied species was compared with the current knowledge on their geographic distribution. While the models overlap mostly in the regions from which those orchids have been reported, we found several incongruences between ENM outcomes and floristic data. Smaller islands of Melanesia and Polynesia were indicated in the analysis as regions suitable for *Apostasia odorata, A. wallichii* and *Neuwiedia zollingeri*; however, we did not find any information on their occurrence in those areas. Additionally, we did not confirm the existence of *Apostasia nuda* or *Neuwiedia borneensis* in New Guinea, *Neuwiedia zollingeri* in the Philippines, or *Apostasia odorata* in Australia. Populations of *Apostasia nuda* were reported, *inter alia*, from Bangladesh, India, Cambodia and Vietnam, but ENM analysis did not indicate continental south-east Asia as suitable for this species. This may be a result of the lack of occurrence data from this area included in the input dataset, and it would suggest that the continental populations grow in different climatic conditions from those distributed along the islands and in the Malay Peninsula.

The most probable reason for the inconsistencies between the potential and recognized geographical ranges of those species is the geographical and/or climatic barriers that prevent migration to suitable habitats across the sea (*Hosner et al., 2014*). On the other hand, we cannot exclude the extinction of some populations in their refugia localized on small islands after deglaciation and sea level rising (*Dávalos & Russell, 2012*). In this situation, there would be no progenitors in some parts of the potential ranges of the studied species. It should also be emphasized that the floristic data from numerous island groups are relatively obscure, and the populations of Apostasioideae representatives may be found in those regions in the future.

## Niche conservatism and evolution of climatic tolerance

The niche conservatism of Apostasioideae is relatively high, and the tolerances of the studied climatic variables of the representatives of this group are rather narrow. The only species which may survive in a broader range of climatic factors is *Apostasia odorata*. Our studies indicated that the distribution of both *Apostasia* and *Neuwiedia* is limited by the same variable—the mean diurnal range. Moreover, within the examined orchids the overlap of their potential ranges is high, and in some cases representatives of *Apostasia* shares more habitats with *Neuwiedia* than with other congeners.

The reconstructed evolution of climatic tolerances of the studied species indicated that after the initial divergence of separated lineages of *Apostasia* and *Neuwiedia* the differentiation of the climatic niches within both genera began. Separation between the two genera occurred approximately 43-41 Ma ago (*Gustafsson, Verola & Antonelli, 2010*) while the diversification of currently recognized species started around 21 to 10 Ma ago (*Guo et al., 2012*). This pattern was also observed in other flowering plants, e.g., palaeotropical genus *Pseuduvaria* (Annonaceae; *Su & Saunders, 2009*). In the time of divergence of Apostasioideae from other orchids, about 45 Ma ago (*Hall, 2002*) Australia began to move rapidly northward. Subduction of the Australian plate resumed beneath Indonesia, causing widespread volcanism at the active margin and producing chains of the Sunda Arc stretched from Sumatra, through Java and the north arm of Sulawesi, and continued into the western Pacific to join the Philippines-Halmahera Arc (*Lohman et al., 2011*). This island chain could be colonized during several events and as proven by *Tänzler et al. (2016)* the *in situ* diversification on different segments of the Sunda Arc occurred. The speciation within studied genera was most probably a consequence of the collision of Sahul and Sunda in the mid-Miocene that resulted also in the emergence of much of the existing land east of Wallace's Line (*Morley, 1998*). Moreover, a second collision event initiated in western New Guinea, creating the Central Range orogeny (*Cloos et al., 2005*; *Baldwin, Fitzgerald & Webb, 2012*).

The niche conservatism within Orchidaceae remains poorly recognized and previous studies were focused mostly on invasive species (e.g., *Kolanowska, 2013*; *Kolanowska & Konowalik, 2014*). The degree to which plants and animals retain their ancestral ecological traits plays essential role in speciation; however, in many orchids this pattern may be misinterpreted and the phylogenetic niche conservatism signal may be disturbed by other factors promoting evolution of new species, especially the role of pollinators. As

assumed by *Ramírez et al. (2007)*, pollination in Apostasioideae must be accidental and the speciation within this group was apparently not related to pollinator specificity. For this reason, apostasioid orchids are ideal model for testing niche conservatism using ENM and molecular data. While the revealed pattern cannot be extrapolated into other Orchidaceae, it brings important information on the ecological differentiation in the earliest stage of orchid evolution.

## ACKNOWLEDGEMENTS

The curators and staff of the cited herbaria are thanked for access to their collections. We are grateful to Sławomir Nowak for providing occurrence records of apostasioid orchids from New Guinea and adjacent areas.

We acknowledge the World Climate Research Programme's Working Group on Coupled Modelling, which is responsible for CMIP, and we thank National Center for Atmospheric Research for producing and making available their model output. For CMIP, the US Department of Energy's Program for Climate Model Diagnosis and Intercomparison provides coordinating support and led development of software infrastructure in partnership with the Global Organization for Earth System Science Portals.

### Funding

The study was financed by the Faculty of Biology, University of Gdańsk (grant nr 538-L150-B583-14). This research received support from the SYNTHESYS Project (http://www.synthesys.info/) which is financed by the European Community Research Infrastructure Action under the FP7 "Capacities" Program (GB-TAF-2445) and from the grant nr 14-36098G of the Grantová agentura České republiky (GA ČR). The funders had no role in study design, data collection and analysis, decision to publish, or preparation of the manuscript.

### Grant Disclosures

The following grant information was disclosed by the authors:
Faculty of Biology, University of Gdańsk: 538-L150-B583-14.
FP7 "Capacities" Program (GB-TAF-2445).
Grantová agentura České republiky (GA ČR): 14-36098G.

### Competing Interests

The authors declare there are no competing interests.

### Author Contributions

- Marta Kolanowska and Kamil Konowalik conceived and designed the experiments, performed the experiments, analyzed the data, contributed reagents/materials/analysis tools, wrote the paper, prepared figures and/or tables, reviewed drafts of the paper.
- Katarzyna Mystkowska performed the experiments, contributed reagents/materials/-analysis tools, prepared figures and/or tables, reviewed drafts of the paper.

- Marta Kras performed the experiments, analyzed the data, contributed reagents/materials/analysis tools, wrote the paper, reviewed drafts of the paper.
- Magdalena Dudek performed the experiments, reviewed drafts of the paper.

## Data Availability

The raw data has been supplied as a Tables S1 and S2.

## Supplemental Information

Supplemental information for this article can be found online at http://dx.doi.org/10.7717/peerj.2384#supplemental-information.

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
