# Peer review of "Evolution of the climatic tolerance and postglacial range changes of the most primitive orchids (Apostasioideae) within Sundaland, Wallacea and Sahul"

_PeerJ, doi:10.7717/peerj.2384_

## Round 0.1 · original submission · Major Revisions

This paper has now been seen by three reviewers. As can be seen from their reports, two reviewers feel that the paper needs (much) more work before it can be considered for publication. I agree with the reviewers that, in general, the experimental design is inadequately described and that the paper lacks clear questions, hypotheses, or conclusions. In general, much more information should be provided on the why and how of this study. Therefore, I would suggest the authors to submit a revised version in which they have carefully considered all comments raised by the reviewers, after which the paper might ultimately be considered for publication.

Reviewer 1 ·

Basic reporting

This manuscript agree with most of these requirements. It would be better to polish the language.

Experimental design

The experimental design is excellent. However, the question need to be more refined.
Your methods follows Hijmans et al. (2005). Hijmans et al (2005) used 30 arc s grid, however, authors use 2.5 arc m grid for analyses. Please clarify.

Validity of the findings

The validity of the findings is good.

Additional comments

This manuscript has discussed spatial dynamics of Apostasioides in both Greater and Lesser Sundland. The major aim of manuscript needs to be refined. And what is the new finding of this manuscript?

Reviewer 2 ·

Basic reporting

Inadequate, see general comments.

Experimental design

Inadequate, see general comments.

Validity of the findings

Unclear, see general comments.

Additional comments

In this manuscript, the authors examine climatic distributions among 6 species of orchids. Overall, I think that the paper is not publishable as currently written. It is not clear what the point of the study is, and most of the methods were not adequately explained. Regardless of the interest level (which seems minimal), there are no clear questions, hypotheses, or conclusions. Simply put, the paper needs to be fully fleshed out (to better explain what was done and why). A more hypothesis-driven approach might be better.

Most species in this subfamily of orchids were not included, but it is not clear why. There is no clear criterion given for why some species were included and some were excluded. It is not even clear why particular localities were included or excluded. For example, it appears that many localities for these species in other Asian countries were excluded, but there is little explanation of this.

A phylogenetic analysis is included, but it is very unclear what was done and why. There is no discussion of what species are included, or where the data even come from. How were the selected genes chosen?

Further, it is not clear why a single model was chosen for all three genes. A more logical approach would be to use PartitionFinder to find the best combination of partitions and models for the combined data.

The authors estimated a phylogenetic tree, but the tree (and its support values) are not directly shown. The authors present analyses based on the relative ages of clades, but there is no discussion of how ages are estimated.

The title of the paper is the “Evolution of the climatic tolerance…” however the climatic tolerances are not actually studied (only climatic distributions) and it is difficult to tell what inferences about evolution (if any) were actually made.

Reviewer 3 ·

Basic reporting

Basically this manuscript is well written. However, introduction and discussion were too much focused on the target species in this study and less wide view of scientific importance in the current manuscript.

Experimental design

As mentioned below, the sample size used to the phylogenetic analysis was quite limited. I know that it is often rather common in phylogenetic study. However, as this study covered huge area of species range by ENM, it would be better to consider the bias of the fact that intra-species genetic variation cannot be evaluated in this study.
PNO should be explained in more detail.

Validity of the findings

As ENM could provide robust result in this study, validity of the findings in this study might meet our standards. However, more detail information would be required especially in introduction and discussion (please see below).

Additional comments

The authors estimated the location of possible refugia of six Apostasioideae representatives using ENM and PNO. The purpose is clear and manuscript was easy to read. Especially, this study would provide new insight of glacial episode of plant species in this area. Thus, this study may be able to be accepted in PeerJ given the following questions are appropriately responded. I have two major comments as follows.

1. Time scale
In L45-56, the authors described the geological history of the continent and Islands in the examined area with time scale of million years. However, the topic moved to the Quaternary in relation to glacial periods from L57 and there is some "temporal gap" here. Moreover, the time scale of “during glaciation” in L59 was not clear. Was it mean during the LGM? I would like to see more detail of the relationship between Apostasioideae and ice ages in these area. There are many papers of phylogeography in several species types in this area and those may be helpful to make this part more fruitful. In addition, time scale of the LGM should be shown at the first description.

2. Phylogenetic analysis and PNO.
If I understood correctly, authors combined the data from bi-parental inherited nuclear DNA (ITS) and maternal inherited chloroplast DNA (matK and trnL) to the phylogenetic analysis. I am wondering if this treatment might mask some genetic signal related to PNO. It would be better to check the result to work on phylogenetic analysis and PNO in each of two genomes. Moreover, according to Table S1, the authors examined only one individual of each of three regions (ITS, matK and trnL) in 6 species. I would like to know the bias by using genetic variation of one individual to discuss the selection pressure over the species range based on ENM. I suppose that there is intra-species genetic variation in the examined regions when more samples are examined in each species. In this case, we need to consider the bias. Regarding PNO, what did “relative age” mean
In Fig6? Probably more detail description would be needed to M&M and the results of PNO.

3. Comparative biogeography/phylogeography
Discussion (as well as introduction) was too much focused on Apostasioideae and it may be attractive only for limited researcher and readers. It would be better to make this study from wider scientific points of view. As mentioned above, comparative biogeography/phylogeography approach would be one of the solution.

---

## Round 0.2 · accepted · Accept

The authors have done their utmost to address all comments of the reviewers and this paper can now be accepted for publication in PeerJ.

Reviewer 3 ·

Basic reporting

The current manuscript has been much improved and the authors answered well in the rebuttal letter to my comments and questions.
I think the current manuscript is suitable to be accepted in PeerJ.

Experimental design

This revised ms showed clear experimental design.

Validity of the findings

The validity of this study in current manuscript is high enough to be published.

Additional comments

Again, the current manuscript was revised well and can be accepted by PeerJ.